# Identification of Novel Umami Peptides from Yak Bone Collagen and Mechanism Exploration Through *In Silico* Discovery, Molecular Docking, and Electronic Tongue

**DOI:** 10.3390/foods14234057

**Published:** 2025-11-26

**Authors:** Yimeng Mei, Xiaoli Wu, Ruoyu Xie, Yulong Wu, Hongying Du, Wenxuan Chen, Jun Hu, Ke Zhao, Runfang Guo, Jin Zhang

**Affiliations:** 1College of Food Science and Technology, Hebei Agricultural University, Baoding 071001, China; 18532399615@163.com; 2State Key Laboratory for Quality and Safety of Agro-Products, Zhejiang Key Laboratory of Intelligent Food Logistic and Processing, Institute of Food Science, Zhejiang Academy of Agricultural Sciences, Hangzhou 310021, China; 3Department of Food Science and Engineering, College of Light Industry and Food Engineering, Nanjing Forestry University, Nanjing 210037, China

**Keywords:** umami peptide, yak bone, collagen, molecular docking, *in silico* screening, electronic tongue, T1R1/T1R3 receptor

## Abstract

Umami peptides were screened and identified from yak bone collagen for the first time by *in silico* analysis, molecular docking, and electronic tongue. Twenty proteases with known cleavage sites were used for the simulated proteolysis, and results indicated that “pepsin + papain” was the optimal enzymatic strategy for yak bone collagen to generate peptides with potential umami taste. Moreover, 82 novel unreported peptides with umami taste were found from the simulated hydrolysate, among which 9 peptides exhibited high binding affinity with the T1R1/T1R3 receptor (both -CDOCKER energy and CDOCKER interaction energy > 40 kcal/mol) via molecular docking. Subsequently, six novel umami peptides were identified through sensory evaluation and electronic tongue analysis, including VY, VM, SL, SN, VN, and IS (umami sensory score > 5). These peptides were also *in silico* characterized with high hydrophobicity, good water solubility, non-toxicity, non-allergenicity, good intestinal absorption, and good oral bioavailability. Furthermore, the identified peptides could bind with the key residues (such as HIS281 and LEU304) within the Venus flytrap domain of the T1R3 subunit of receptor T1R1/T1R3 through hydrogen bonds and electrostatic attractions to generate umami perception. This study revealed the mechanism of umami peptides identified from yak bone collagen and provides a novel approach for the development of umami peptides from animal sources.

## 1. Introduction

Yak (*Bos mutus*), hailed as the “ship of the highland”, is a characteristic animal mainly living in the China Qinghai–Tibet Plateau with a height > 3000 and oxygen content < 75% (on a plain basis) [1]. As an agro-product with geographical indication status, the yak industry has received strong support from local governments, which promote large-scale breeding and industrial processing to boost local economic income [2]. These past few years, as the tourism and logistics sectors have grown quickly, the need for chilled yak beef served at meals has kept increasing, and ready-to-eat items meant for travelers have been in short supply [3]. However, yak bone is usually discarded as waste during processing. However, China’s yak bone production approaches 90,000 tons, yet utilization rates remain below 10%. Each yak slaughter yields substantial quantities of fresh bones, most of which are discarded after minimal use for making soup or producing bone meal. This approach not only squanders resources but also adds to environmental pollution [4]. Research indicates that yak bones are packed with high-quality protein, which is essential for human nutrition, and offer significant utilization value as an ideal starting material for preparing bioactive peptides [5]. For instance, feeding mice hydrolyzed yak bone collagen can shift their gut microbiota and boost short-chain fatty acid levels [5], ultrafine bone powder helps ease osteoporosis [6], and anti-inflammatory peptides can be extracted from yak bones [7].

Umami imparts a rich, appealing flavor to food and enhances its overall palatability. Monosodium glutamate (MSG) remains the most commonly used umami additive [8]. However, due to its relatively high sodium content, an excessive consumption of MSG may lead to hypertension, hyperuricemia, and other cardiovascular diseases [9]. Studies showed that some free L-amino acids and peptides can deliver or intensify the favorable umami perception [10]. The emergence of umami peptides could perfectly address the shortcomings of MSG, and thereby, they have become a prospective alternative to MSG applied in the food industry. Derived from natural proteins and released through gentle enzymatic hydrolysis, this peptide imparts a rich, lingering “fifth taste” to broths and sauces, without dosage concerns. As a widely favored umami source, it demonstrates unique development potential [11,12]. The mechanism of taste perception involves umami peptides latching onto heterodimeric T1R1/T1R3 receptors on the tongue. This attachment makes the receptor’s Venus Flytrap Domain (VFT) snap shut, triggering the umami signal. This activates T1R1/T1R3, thereby triggering the brain to generate pleasurable umami signals [13,14]. Most of the identified umami peptides have been reported to be isolated from animal proteins such as chicken, fish, and clams [15,16,17]. In contrast to other sources of proteins, the mining of novel antioxidant peptides from yak bone has not yet been reported.

For a long time, protein extraction has primarily relied on enzymatic hydrolysis technology. Furthermore, the identification of new bioactive peptides relies on sensory analysis following many precise separation and purification steps, a process that quickly racks up both bills and hours [18]. Lately, however, researchers have turned to virtual screening as a faster and cheaper way to uncover these beneficial molecules, successfully hunting for umami-enhancing peptides [19], DPP-IV inhibitor peptides [20], and antioxidant peptides [21]. Moreover, molecular docking technology has been extensively applied to establish suitable interactions between ligands and receptors, thereby accelerating the drug discovery process while simultaneously revealing potential mechanisms of action [19]. Compared to traditional methods, identifying peptides using computational analysis and molecular docking technology not only offers higher efficiency and more reliable results but also significantly reduces time and costs.

Therefore, the objective of the present study was to screen, identify, and characterize novel umami peptides derived from yak bone collagen through a combined approach of *in silico* analysis and molecular docking. By mapping out how the peptides interact with the T1R1/T1R3 receptor, we shortlisted the most promising candidates. Their savory character was then double-checked with a tasting panel and an electronic tongue, giving us the final set of umami peptides.

## 2. Materials and Methods

### 2.1. Materials

Citric acid, sucrose, quinine sulfate dihydrate, sodium chloride, and monosodium glutamate were purchased from Sigma-Aldrich Co., Ltd. (St. Louis, MO, USA) and China National Medicines Co., Ltd. (Beijing, China), respectively. All chemicals and reagents used were of food grade. Molecular docking analysis was performed using Discovery Studio V2019 (DS2019) software (Dassault Systèmes Biovia, San Diego, CA, USA). All synthesized peptides (purity > 95%) were provided by Shanghai Dechi Biosciences Co., Ltd. (Shanghai, China).

### 2.2. Comparison of Collagen Sequences from Yak Bone and Bovine Bone

Of the twenty-nine collagen variants so far cataloged, type I collagen predominates, comprising 90–95% of the organic constituents present in bone tissue [4]. Therefore, yak bone type I collagen was selected as the research subject. The sequences of yak bone collagen α1(I)-chain and α2(I)-chain were obtained from the NCBI (https://www.ncbi.nlm.nih.gov/ accessed on 15 January 2025) website under the accession numbers ELR60286.1 and ELR46121.1, respectively. α1(I)-chain (sequence number: P02453.3) and α2(I)-chain (sequence number: P02465.2) of collagen from bovine bone were selected. Sequence comparison of yak bone collagen (α1(I)-chain and α2(I)-chain) and bovine bone collagen (α1(I)-chain and α2(I)-chain) was performed using the DNAMAN V10 software.

### 2.3. Simulated Proteolysis and Peptide Screening

Based on the method of Lin et al. [22] and Lee et al. [13], with slight modification, we simulated the hydrolysis of yak bone proteins and predicted the cleavage sites by using the “Enzyme(s) Action” tool in the BIOPEP-UWM database (accessed on 16 January 2025). 20 representative industrial proteases, including chymotrypsin A (EC 3.4.21.1), chymotrypsin C (EC 3.4.21.2), metridin (EC 3.4.21.3), pancreatic elastase (EC 3.4.21.36), leukocyte elastase (EC 3.4.21.37), chymase (EC 3.4.21.39), trypsin(EC 3.4.21.4), subtilisin (EC 3.4.21.62), proteinase K (EC 3.4.21.64), plasmin (EC 3.4.21.7), pancreatic elastase II (EC 3.4.21.71), oligopeptidase B (EC 3.4.21.83), proteinase P1 (lactocepin) (EC 3.4.21.96), papain (EC 3.4.22.2), ficin (EC 3.4.22.3), stem bromelain (EC 3.4.22.32), calpain 2 (EC 3.4.22.53), pepsin (pH = 1.3) (EC 3.4.23.1), thermolysin (EC 3.4.24.27), and coccolysin (EC 3.4.24.30), were used for virtual enzymatic hydrolysis of yak bone collagen. The frequency of fragments with umami (*A_umami_*) was obtained from Equation (1), and the theoretical degree of hydrolysis (*DH*) was calculated from Equation (2) [23]:(1)$$A_{\mathrm{umami}}=\frac{a_{\mathrm{umami}}}{N}$$(2)$$DH=\frac{d}{D}\times 100\%$$
where *A_umami_* is the frequency of umami fragments in the protein sequence, *a_umami_* is the number of umami fragments in this sequence, and *N* represents the number of total amino acid residues in this sequence; *DH* is the predicted degree of hydrolysis for this sequence, *d* is the number of hydrolyzed peptide bonds in this sequence, and *D* is the total number of peptide bonds in this sequence. The data of *a_umami_*, *N*, *d*, and *D* were all sourced from the BIOPEP-UWM database.

Then, five proteases with both high *A_umami_* and high *DH* were selected for co-enzymatic digestion (combination of two or three proteases) to obtain the maximum number of potential umami peptide fragments. All generated sequences were screened against the BIOPEP-UWM repository and published literature; only peptides without prior umami annotation were advanced to the next stage.

### 2.4. Homologous Modeling of T1R1/T1R3 Receptor

The best available structure of the heterodimer T1R1/T1R3 was established via homology modeling. The homology modeling workflow is as follows: The amino acid sequences of the umami heterodimer T1R1/T1R3 (Q7RTX1 and Q7RTX0) were obtained from UniProtKB (https://www.uniprot.org/, accessed on 17 January 2025). Subsequently, PDB ID: 1EWK was selected as the template. Homology modeling was performed using SWISS-MODEL (https://swissmodel.expasy.org/interactive, accessed on 17 January 2025) based on template 1EWK and the target sequences [24]. After being imported into Discovery Studio 2019, the starting homology model was refined by a short kinetic run and then checked for structural quality using a Ramachandran plot and Profiles-3D scores [25].

### 2.5. Molecular Docking Between Peptides and T1R1/T1R3

We ran docking simulations between the potential umami peptides and the T1R1/T1R3 receptor in DS2019, adopting the procedure outlined by Jiang et al. [9] with only minor tweaks. Before docking, the T1R1/T1R3 structure was imported into DS2019 and pre-processed with the “Prepare Protein” module: crystallographic waters were stripped, hydrogens were added, and the entire receptor was subjected to energy minimization. According to the research report by Jiang et al. [9], the docking pockets were defined as X: 33.883, Y: −2.30462, Z: 35.6531, and radius: 20 Å. Simultaneously, all potential umami peptides were constructed using the “Protein Preparation” module in DS2019 software and underwent preprocessing such as energy minimization, followed by semi-flexible docking via the CDOCKER protocol [20]. Specifically, the receptor (T1R1/T1R3) was set to be rigid, and the ligand (candidate umami peptide) was set to be flexible. For each ligand–receptor pair, the top pose was scored using the -CDOCKER energy (-CE) and -CDOCKER interaction energy (-CIE) as the evaluation criteria. Peptides showing the highest -CE and -CIE were considered as the most probable novel umami peptides and were evaluated next. Finally, the three-dimensional (3D) and two-dimensional (2D) diagrams of binding sites and non-binding interaction patterns of the umami peptides were obtained.

### 2.6. In Silico Analyses of Physicochemical Properties

Potential umami peptides were screened based on the method of Zhang et al. [20] (with minor modifications). The PepDraw server (https://webs.iiitd.edu.in/raghava/toxinpred/, accessed on 19 January 2025) was employed to obtain the peptides’ molecular weight, hydrophobic character, isoelectric point, and net charge. Peptide toxicity was assessed via the ToxinPred tool (https://webs.iiitd.edu.in/raghava/toxinpred/, accessed on 20 January 2025). Sensitization potential was assessed using the AllergenFP 1.0 tool (https://ddg-pharmfac.net/AllergenFP/, accessed on 20 January 2025). Peptide water solubility was predicted via the SwissAMDE tool (http://www.swissadme.ch/index.php, accessed on 21 January 2025). Additionally, the human Intestinal Absorption (HIA) Probability and human Oral Bioavailability (HOB) were evaluated using the iDrug platform (https://webs.iiitd.edu.in/raghava/toxinpred/, accessed on 21 January 2025). Ultimately, only peptides meeting the following criteria were included in subsequent studies: non-toxicity, non-allergenicity, a proportion of umami amino acid residues > 0, water solubility > 0, and molecular docking values for both -CE and -CIE > 40 kcal/mol.

### 2.7. Solid-Phase Synthesis of Umami Peptide Candidates

As described by Zhang et al. [20], peptides were assembled by Fmoc solid-phase synthesis. Crude products were purified to > 95% (*w*/*w*) on an Agilent 1260 HPLC with a Sinochrom ODS-BP column (250 × 4.6 mm, 5 µm) at 1 mL min^−1^, detecting at 220 nm. A linear gradient of 10 → 35% A (0.1% TFA in CH_3_CN) over 25 min, then 0.1 min ramp to 100% A, was used. Accurate masses were confirmed on an AB SCIEX 5600-plus mass spectrometer (Shanghai, China).

### 2.8. Identification of Umami Peptides by Sensory Evaluation

Referring to the method of Wang et al. [26] with minor modifications, the sensory evaluation panel was composed of eight laboratory personnel—four male and four female—aged between 20 and 35 years. All participants were in good health, had no history of smoking or taste-related disorders, and provided informed consent prior to taking part in the sensory analysis conducted in this study. The five basic standard solutions could be easily identified after professional training: citric acid (0.12 g/100 g) for sourness, glucose (1.5 g/100 g) for sweetness, L-leucine (0.15 g/100 g) for bitterness, table salt (0.35 g/100 g) for saltiness, and monosodium glutamate (0.35 g/100 g) for umami. Tasting was performed on a 10-point scale: 1–3 barely detectable, 4–7 noticeable, 8–10 intense. Judges rinsed with drinking water before and after each sip and rested for one minute before the next sample.

### 2.9. Electronic Tongue Measurement of Identified Umami Peptides

The taste properties of the synthetic peptides were measured using an electronic tongue, referring to Zhao et al. [27] with minor modifications. An Ag/AgCl reference electrode was employed to record, for every sensor, the full amplitude of its current response to each applied taste stimulus. Prior to the commencement of taste assessment, the sensor membranes were left to equilibrate in the standardized flavor solution until a constant and stable potential was achieved. After each sample measurement, the electrodes were subjected to an electrochemical rinsing protocol with ultrapure water to ensure complete removal of any residual analytes and thus prevent cross-contamination before the next sample was introduced. To guarantee data reliability and stability, each sample underwent four consecutive recordings; the first reading was dropped, and the average of the last three was used for all further calculations.

### 2.10. Statistical Analysis

All experiments were performed in triplicate, and results are shown as means ± standard deviation. The technical analysis for each biological sample was also performed in triplicate. Figures were drawn with the Origin V2022 software (Origin-Lab, Northampton, NC, USA) and Microsoft PowerPoint 2016, while Tables were made with Microsoft Excel 2016. Statistical analysis was performed using SPSS 25 software (SPSS software, Chicago, IL, USA) employing one-way ANOVA or Student’s *t*-test and confirmed statistically significant when *p* < 0.05.

## 3. Results and Discussion

### 3.1. Sequence Alignment of Bone Collagens from Yak and Bovine

Homologous proteins are generally considered to be a class of proteins derived from peptides with similar biological functions. Partial sequences of the α1(I) and α2(I) chains derived from yak bone collagen are displayed in Figure 1a,b, respectively, alongside the corresponding segments of bovine bone collagen for direct comparison. The α1(I) chain of yak bone collagen comprises 1463 amino acid residues, whereas its bovine counterpart contains 1459 residues; similarly, the α2(I) chain of yak bone collagen spans 1364 residues, while that of bovine bone collagen spans 1366 residues. In addition, the sequences of α1(I) and α2(I)-chains of yak and bovine bone collagen were highly homologous (99.73% and 99.19%, respectively). Wang et al. [28] reportedly extracted a new umami peptide from bovine bone that can replace NaCl. Xu et al. [29] demonstrated that bovine bone protein hydrolysate is able, through the Maillard reaction, to deliver characteristic umami components to food matrices, thereby augmenting the overall flavor and aromatic attributes of the final products. Owing to the pronounced sequence homology between bovine and yak bone collagen, the latter is equally suited to serve as a robust precursor for the generation of diverse bioactive peptides.

### 3.2. Simulated Proteolysis of Yak Bone Collagen

Although differences still exist compared with practical biological enzymolysis, *in silico* simulated enzymolysis has been reported with increasing predictive accuracy and is widely applied in peptide exploration. Huang et al. [30] found that two xanthine oxidase inhibitory peptides identified from fire hemp seed hydrolysates contain three released peptide sequences through *in silico* simulated enzymolysis. Gao et al. [31] reported that the key results from *in silico* and in vitro simulated gastrointestinal digestion are highly consistent, particularly demonstrating a complete alignment in functional activity trends and molecular mechanism validation. Preparation of peptides by *in silico* simulation of protein hydrolysis not only reduces the cost of the project but also effectively disposes of the waste. In this study, *in silico* protein hydrolysis of yak bone collagen (α1(I)-chain and α2(I)-chain) sequences was performed using 20 representative proteases with different known cleavage sites from industrial use or the gastrointestinal tract. In addition, *A_umami_* and theoretical *DH* values were calculated to assess the enzymatic hydrolysis effects, which are exhibited in Figure 2a,b. It is illustrated that both α1(I)-chain and α2(I)-chain sequences showed different *A_umami_* and theoretical *DH* values after the hydrolysis under a single protease. Furthermore, the *A_umami_* value seems overall positively correlated with the theoretical *DH* value, indicating that a relatively higher degree of hydrolysis could promote the release of umami peptides under the condition of single proteolysis.

According to *A_umami_* as well as theoretical *DH*, five proteases were selected for combinatorial digestion, including pepsin, papain, pancreatic elastase, stem pineapple protease, and calpain 2. In addition, the order of action of these proteases was pepsin (*A_umam__i_* = 0.0833–0.0846, *DH* = 65.3–65.5%) > pancreatic elastase (*A_umam__i_* = 0.0313–0.0242, *DH* = 51.1–54.3%) > stem bromelain (*A_umam__i_* = 0.0306–0.0270, *DH* = 54.6–57.7%) > calpain 2 (*A_umam__i_* = 0.024–0.0185, *DH* = 50.7–52.0%) > papain (*A_umam__i_* = 0.0193–0.0206, *DH* = 45.4–7.3%) (Figure 2a,b). In agreement with this result, Zhang et al. [11] found that the Huangjiu lee hydrolysates from pepsin treatment possessed significantly higher yield of total peptides than those hydrolyzed by papain, alkaline protease, and neutral protease. Moreover, it has been reported that the cleavage sites of pepsin are in a wide range [32], such as the residues F, L, G, Y, A, E, Q, and T (C-terminus) (data from the BIOPEP-UWM database, accessed on 18 January 2025). In addition, the pancreatic elastase usually tends to cleave sequences into short peptides [33], and its preferential cutting sites include A, G, V, L, I, Y, S, and T (C-terminus) (data from the BIOPEP-UWM database, accessed on 18 January 2025). Yak bone collagen sequences are rich in A, G, L, Y, and T residues, providing a large number of cleavage sites for pepsin and pancreatic elastase, which may be the main reason why they have the highest *A_umami_* and *DH* values in the results of hydrolysis by all single proteases.

Given that pepsin was significantly better than the other four proteases, pepsin was chosen to be combined with the other proteases for enzyme digestion (a combination of two or three proteases) to obtain the maximum number of potential umami peptide fragments. The results of combinatorial protein hydrolysis of α1(I) and α2(I)-chains of yak bone collagen are shown in Figure 2c,d, respectively. It is clear from Figure 2c,d that combined protease hydrolysis usually increased the *DH* value of yak bone collagen compared to single protease hydrolysis. However, when the *DH* value exceeded about 75%, the inhibitory effect of *A_umami_* diminished with a further increase in *DH* value. It has been shown that excessive hydrolysis produces more free amino acids, leading to a reduction in the number of released peptide fragments [34,35]. The results showed that the combined protein hydrolysis of pepsin and papain under appropriate conditions was the best enzymatic method for the preparation of umami peptides from yak bone collagen (*A_umami_* = 0.0879–0.0882, *DH* = 74.1–74.6%). Therefore, we further analyzed the hydrolysis of “pepsin + papain” to explore novel umami peptides.

### 3.3. Release Profile and Screening of Peptides

Figure 2e,f show the amino acid and peptide profiles of yak bone collagen α1(I) and α2(I)-chains, respectively, released by *in silico* protein hydrolysis by “pepsin + papain”. Seventy-seven peptides of 2–4 residues were released from the α1(I)-chain of yak bone collagen (Figure 2e), and four of these dipeptides were identified as umami peptides in the BIOPEP-UWM database, including VD, VE, VG, and DA. 18 umami peptides of 2–3 residues were also reported in previous peer-reviewed studies, such as VT [36], SE [37], and IPP [38]. Thus, 54 dipeptides, tripeptides, and tetrapeptides were screened after excluding the above peptides. Yak bone collagen α2(I)-chain also released 64 peptides of 2–4 amino acid residues (Figure 2f). Among them, VE, VG, and VD were identified as umami peptides in the BIOPEP-UWM database. In addition, 11 dipeptides and tripeptides were found to be umami peptides in peer-reviewed articles, such as SD [37], PN [39], and PSG [40]. Therefore, after removing the peptides reported above, we selected 50 peptides (dipeptides to tetrapeptides) as potential novel umami peptides. Overall, a total of 82 peptides were selected after removing the repetitive fragments (Figure 2e,f).

### 3.4. Molecular Docking of Screened Peptides with T1R1/T1R3 Receptor

Taste receptor T1R1/T1R3 is a heterodimer whose extracellular portion contains a flytrap-like structural domain (Venus Flytrap Domain, VFT), which serves as a binding site for taste ligands. Umami substances are able to enter and bind to the VFT, which triggers conformational changes and initiates downstream signaling, ultimately resulting in umami perception [41]. In previous studies, -CIE and -CE are key parameters widely used to assess target ligand or receptor interactions [42,43,44]. The -CIE is commonly used to reflect the stability and affinity of binding. The –CE binding energy encompasses not only the direct interaction energy between ligand and receptor but also incorporates any strain energy that may arise within the ligand as a consequence of the binding event [45]. Therefore, -CIE and -CE are important for screening potentially active molecules. Before docking, a homology model for T1R1/T1R3 was constructed, and its reliability was verified. Figure 3a shows the finished homology model: T1R1 sits on the left and T1R3 on the right. According to the Ramachandran plot provided in Figure 3b and the residue distribution data summarized in Figure 3c, 98% of the amino acid residues were in the reasonable region (88.5% in the allowed region and 9.5% in the marginal region) and 2% in the disallowed region. Therefore, the model constructed by this homology simulation is reasonable. The validation score (Profiles-3D) assessment (Figure 3c) showed that 83.39% of the residues had an average 3D/1D score of 0.2, so the umami receptor model is reasonable.

Table 1 shows the physicochemical properties and biological potential of the umami peptide candidates. In analyzing 82 peptides from an initial screening, we found that 16 of these peptides (ranging from dipeptides to tripeptides) did not contain umami amino acids. We sense umami when taste receptors meet a mix of savory triggers, including specific taste-active peptides, Maillard-reaction products, free amino acids, and 5′-nucleotides [46,47]. According to Liu et al. [48] and Zhang et al. [49], 10 umami amino acids have been identified, which include Asp (D), Glu (E), Gly (G), Tyr (Y), Ala (A), Phe (F), His (H), Thr (T), Val (V) and Ser (S). These results indicate that the umami character of peptides is intimately linked to their amino acid makeup and that peptides lacking umami-contributing residues may exhibit altered taste behavior. In addition, toxicity and water solubility, which affect the food application and normal metabolism of peptides, are two important parameters [42,44]. All peptides were predicted to be non-toxic to humans, but seven of them had a water solubility of <0. In addition, 17 of these peptides were potentially allergenic. In conclusion, 44 peptides were finally selected for subsequent studies.

Afterward, the molecular dockings between these peptides and the T1R1/T1R3 receptor were conducted, and the docking energies (-CE) and docking interaction energies (-CIE) are shown in Figure 4. Overall, 9 of these peptides failed to dock with T1R1/T1R3, indicating that they were not potential umami peptides. All the other 35 peptides could successfully interact with the active site of T1R3, which might be due to the fact that T1R1 is closed and T1R3 is open in the model [50]. In addition, the 14 previously reported umami peptides within the released fragments (Figure 2e,f) showed average -CIE and -CE values of 40.92 and 43.03 kcal/mol, respectively, with T1R1/T1R3 (Appendix A). Meanwhile, 9 among the 35 successfully docked peptides showed values of both -CIE and -CE greater than 40 kcal/mol. Therefore, these 9 peptides with both high -CIE and -CE were considered novel umami peptide candidates, including VY, VM, IT, SL, SN, SM, VN, IS, and HN. Additionally, the High-Performance Liquid Chromatography (HPLC) and Mass Spectrometry (MS) reports of these peptides are presented in Appendix A.

### 3.5. Analysis of Physicochemical Properties for Potential Umami Peptides

Table 2 shows the computer-predicted physicochemical properties and biological potential of nine umami peptide candidates. The results indicate that these peptides are all dipeptides with molecular weights (MWs) < 300 Da. Research indicates that umami peptides typically have low molecular weights [51]. 90% of peptides exhibit a pI significantly distant from 7.0 in neutral solutions and carry no net charge, indicating that binding of candidate peptides to T1R1/T1R3 does not depend on electrostatic interactions. Additionally, approximately 90% of candidate peptides exhibit hydrophobicity values below 10 kcal/mol. Some studies suggest that lower hydrophobicity may correlate with umami expression, and Wang et al. [52] also employed lower hydrophobicity as an indicator for evaluating umami peptides. All candidate peptides exhibit an HIA probability of 1 and the human oral bioavailability (HOB) exceeding 60%, indicating their potential for efficient absorption and umami functionality. *In silico* prediction accelerates peptide activity screening through its low-cost, high-throughput advantages, yet remains constrained by issues such as data quality, model interpretability, and insufficient experimental validation. Its practical application still requires breakthroughs. Overall, this group of small-molecule, uncharged, highly absorbable dipeptides holds development potential as novel umami enhancers.

### 3.6. Synthesis and Identification of Umami Peptides by Sensory Evaluation

The results of the sensory evaluation are shown in Figure 5a, where the nine synthetic peptides are rich in flavor, displaying not only umami but also other flavor characteristics such as sour, sweet, salty, and bitter, which is in accordance with the report of Li et al. [53]. Among these, umami and sourness dominate the overall flavor profile of the peptide compounds. Additionally, all synthetic peptides are accompanied by a slightly bitter taste. This may be related to the residual reagents in the synthesis process [54]. All the peptides had similar umami characteristics, with VY and IS having the most pronounced umami with the highest score (7.01), SM, HN, and IT all scored less than 5, with 4.91, 4.82, and 4.82, respectively. In this study, only peptides with umami scores higher than 5 were screened for the next step. The decisive factor for umami perception is not restricted to amino-acid composition; the three-dimensional conformation, together with the presence of umami-residue, hydrophilic and hydrophobic motifs, likewise forms the fundamental prerequisite for eliciting umami [55]. As described by Chang et al. [54], Hydrophobic residues such as Val (V), Leu (L), Ile (I), and Met (M) exhibit a preferential affinity for the umami receptor and thereby make a critical contribution to the expression of umami taste. Acidic amino acids such as Ser, Thr, Ala, and Gly are also believed to contribute to the production of umami and are important umami stimulants. These residues tend to appear in the sequences of umami peptides and in certain amino acids that also carry a sweet note [56]. This property may be associated with the umami flavor displayed by the peptides discovered in the current study.

### 3.7. Electronic Tongue Analysis of Identified Novel Umami Peptides

Figure 5b shows a double-label plot of the e-tongue PCA, with PC1 and PC2 contributing 92.6% and 4.9%, respectively, and together explaining 97.5% of the variance, suggesting that PC1 and PC2 better reflect the flavor of the synthetic peptide. The differences between the different peptides were mainly caused by PC1, with VN, IT, and MSG mainly distributed on the negative axis of PC1, suggesting that the flavor profile of these two peptides is more similar to that of MSG. SM and VY are predominantly distributed on the positive axis of PC1 at a distance from the MSG, which is at variance with the results of the sensory evaluation. Furthermore, Yu et al.’s [57,58] research on electronic tongues also indicates that the taste characteristics of different peptides exhibit significant differences, consistent with our findings. Research findings have shown that human-based sensory assessments provide a complete and direct measure of the perceived intensity of particular attributes. In human perception, interactions may occur between different flavors, which may inhibit or enhance certain flavor perceptions. For example, acidity can, in some cases, reduce the perceived intensity of umami [58]. Therefore, the results of the human sensory evaluation were used as the main reference, while the results of the electronic tongue were used as a complement.

### 3.8. Potential Interactive Mechanism of Identified Peptides with Umami Receptors T1R1/T1R3

Evidence indicates that the T1R3 subunit serves as the principal locus for the docking of umami peptides [50]. Figure 6a shows the binding site of the umami peptides to the T1R1/T1R3 receptor, and the results indicate that all six novel umami peptides are able to access the VFTD binding domain of T1R3. The -CIE order of peptide binding to the receptor was: VY > VM > SL > SN > VN > IS. Lower docking energies indicate a more stable conformation. Results showed VY and IS received the highest umami scores. This suggests that umami intensity cannot be judged solely by the binding energy value, which matches the findings from Zhao et al. [27]. HIS281 and LEU304 showed a high occurrence rate in docking, and this confirmed them as the main binding sites of umami peptides to the T1R1/T1R3 receptor (Figure 6h). As reported by Dang et al. [59], the receptor residues that accommodate umami peptides are predominantly polar; these polar side chains are capable of establishing superior interactions with the peptides by means of hydrogen bonding. In this study, polar amino acids such as HIS, ARG, and SER appeared repeatedly, and these amino acids were shown to play a key role when umami peptides dock with the umami receptor. Yang et al. [60] also highlighted the importance of ASN68, HIS145, and LEU304 in the molecular docking process with the T1R3 receptor. In conclusion, ARG52, ASN68, HIS281, SER104, HIS278, LEU304, HIS145, SER306, and SER66 play key roles in the binding of umami peptides to the receptor in the present study.

Figure 6i shows the non-bonding interaction sites for peptide binding to T1R1/T1R3, and the results indicate that the main non-covalent interaction modes between these peptides and T1R1/T1R3 can be categorized as hydrogen bonds (conventional, carbon, and Pi-Donor Hydrogen Bond) and hydrophobic interactions (Pi-Sigma, Pi-Alkyl, and alkyl). Hydrogen bonds form far more often than other kinds of interactions, such as ASN68, LEU304, SER306, SER66, HIS281, TRP303, VAL309, MET310, ARG52, SER104, HIS145, and PRO313. These residues participate in the interactions of most identified peptides via hydrogen bonds, and this matches existing findings that highlight hydrogen bonds as vital for umami peptides to bind stably to T1R1/T1R3 receptors [17]. Furthermore, the hydrophobic interactions formed between peptide chains and residues (Including ALA280, PRO313, PHE284, VAL277, LEU304, ALA280, ARG52, HIS281, HIS145, and HIS278) of the T1R1/T1R3 taste receptors effectively boost the binding strength between the two, which in turn makes taste perception stronger.

Surface forces between umami peptides and T1R1/T1R3 complexes are illustrated in Figure 7. Prior studies have established that aromatic interactions among aromatic residues such as Phe, Tyr, and Trp represent essential non-covalent interactions that stabilize both protein and peptide architectures. In this study, the electrostatic interaction of protons on the face ring was weaker than the π-electron density on the edge ring of T1R3 and umami. In addition, all six umami peptides formed strong hydrogen bonds, especially on HIS and LEU residues. Additionally, every umami peptide displayed pronounced hydrophilicity, an observation that can be ascribed to the high density of hydrophilic moieties present on both the peptides and the T1R1/T1R3 receptor. In this study, there was a high solvent-accessible surface (SAS) between the umami peptide and T1R3, suggesting their binding through van der Waals interactions. Other surface force types are stronger than the IC and ionizability between T1R3 and umami peptides. This indicates that the influence of these two forces on the interaction between the six umami peptides and T1R3 may be limited, consistent with what was found in [16]. In summary, umami peptides’ binding to receptors and the production of umami are supported by aromatic interactions, H-bonds, hydrophilicity, and SAS.

## 4. Conclusions

In this study, we used simulated hydrolysis and virtual screening techniques combined with sensory evaluation and electronic tongue technology to rapidly screen six novel umami peptides with good binding affinity (-CIE > 40 kcal/mol and -CE > 40 kcal/mol) to T1R1/T1R3 from yak bone type I collagen, namely VY, VM, SL, SN, VN and, IS. These peptides also feature low molecular weight, high hydrophobicity, carry no net charge, good water solubility, non-toxicity, and non-allergenic properties. Sensory evaluation and electronic tongue analysis results revealed that every one of these peptides possessed a good umami profile. Furthermore, during molecular docking, all six peptides successfully inserted into the active site of T1R3 and the key amino acids ARG52, ASN68, HIS281, SER104, LEU304, HIS145, SER306, and SER66, which play an important role in the binding process. All peptides bind to HIS281 and LEU304 of T1R3 via hydrogen bonding. In addition, the aromatic interaction, hydrogen bond, hydrophilicity, and SAS served as the key interaction forces. In comparison to traditional separation methods, this approach cuts down the cycle time noticeably and lowers costs—at the same time, it also confirms that virtual screening is reliable for discovering umami peptides.

## Figures and Tables

**Figure 1 foods-14-04057-f001:**
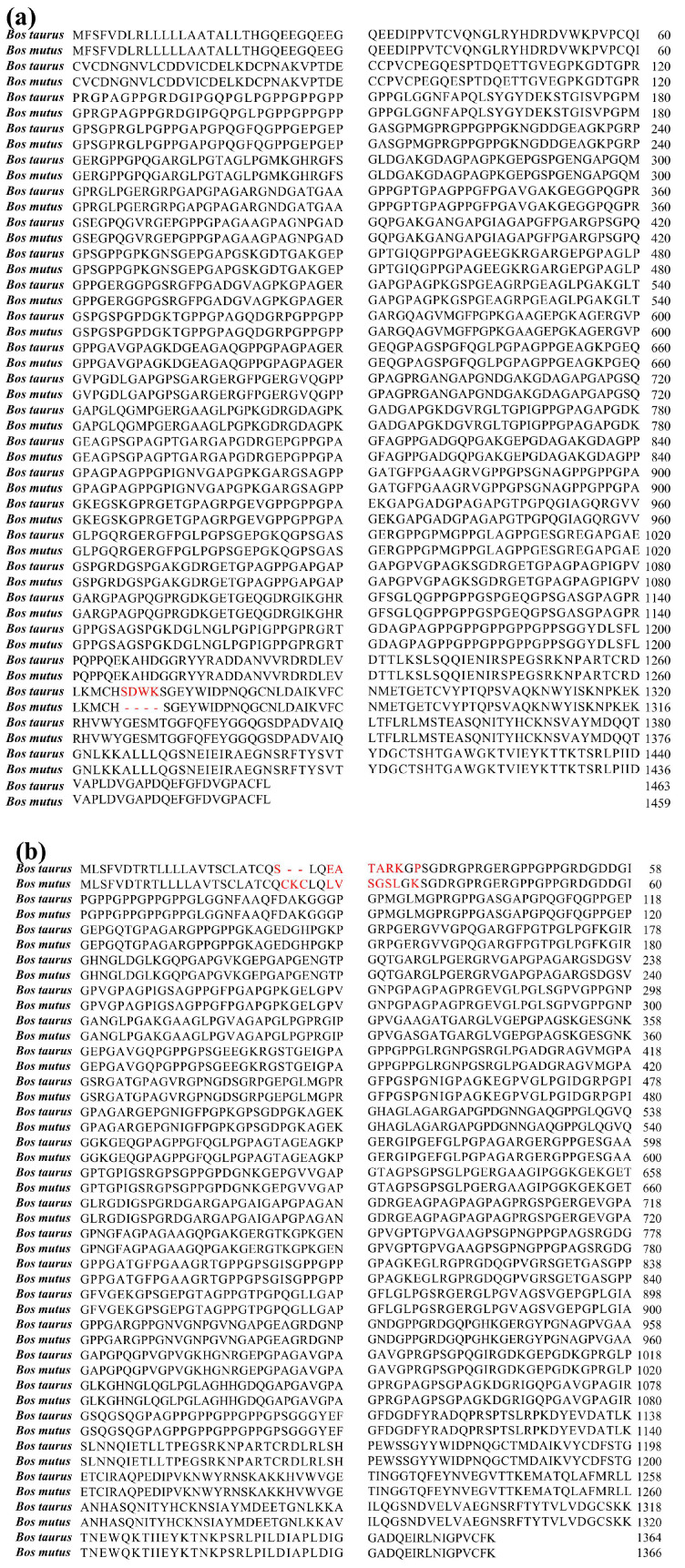
Sequence comparative alignment of yak bone collagen against bovine bone collagen. (**a**), α1(I)-chain sequence of yak bone collagen vs. α1(I)-chain sequence of bovine bone collagen. (**b**), α2(I)-chain sequence of yak bone collagen vs. α2(I)-chain sequence of bovine bone collagen.

**Figure 2 foods-14-04057-f002:**
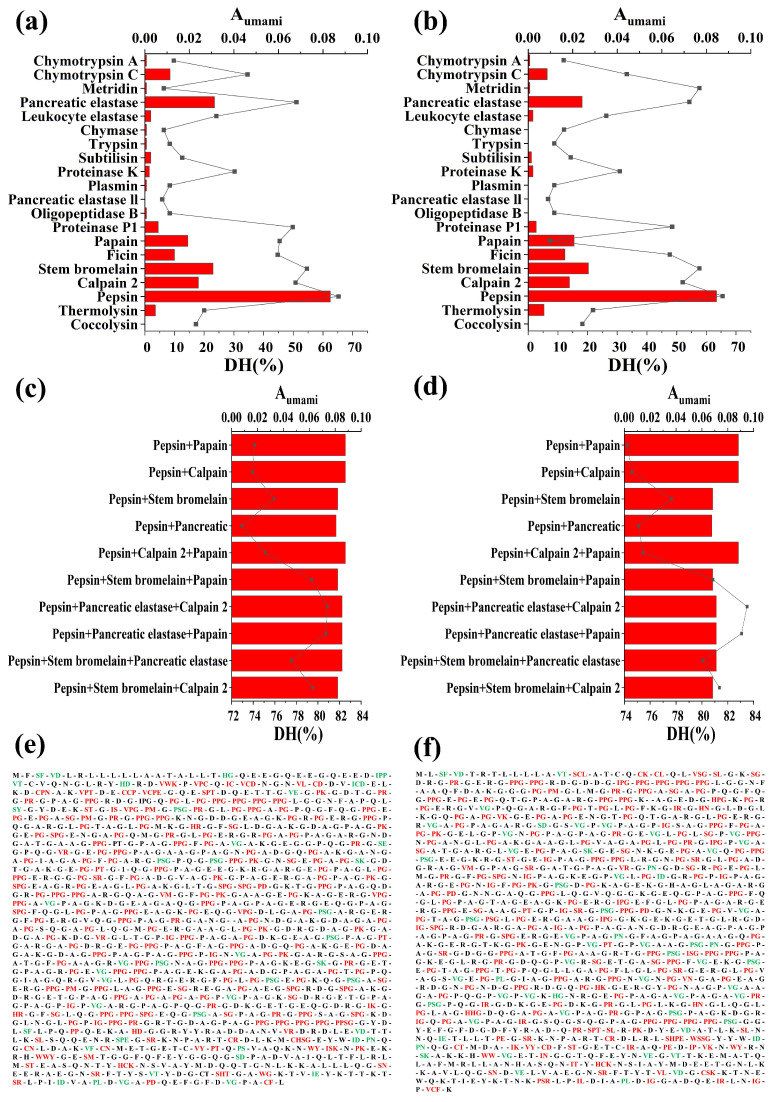
Frequency of umami peptides (*A_umam_*_i_), degree of hydrolysis (*DH*), and release profile of peptide fragments after *in silico* simulated proteolysis of yak bone collagen. (**a**,**b**), *A_umami_* and *DH* of α1(I)-chain (**a**) and α2(I)-chain (**b**) from yak bone collagen subjected to enzymolysis with a single protease, respectively. (**c**,**d**), *A_umami_* and *DH* of α1(I)-chain (**c**) and α2(I)-chain (**d**) from yak bone collagen subjected to enzymolysis with combined proteases, respectively. The *A_umami_* and *DH* values were presented as columns and scatters in (**a**–**d**), respectively. (**e**,**f**), results of simulated enzymolysis for α1(I)-chain (**e**) and α2(I)-chain (**f**) of yak bone collagen, respectively, using optimal combined proteases (pepsin + papain) based on *A_umami_*. The known umami peptide sequences by the BIOPEP-UWM database or previous peer-reviewed reports were exhibited in a green color, whereas peptide sequences with unknown umami taste were shown in a red color.

**Figure 3 foods-14-04057-f003:**
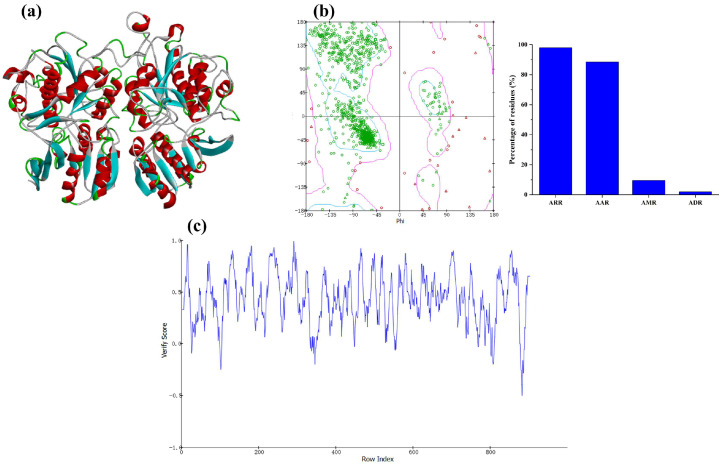
Homology modeling of the T1R1/T1R3 and its reliability. (**a**) Model of T1R1/T1R3. (**b**) Raman map of T1R1/T1R3 and Distribution of amino acid residues in T1R1/T1R3. ARR indicates amino acid residues in the reasonable region. AAR indicates amino acid residues in the allowed region. AMR indicates amino acid residues in the marginal region. ADR indicates amino acid residues in the disallowed region. (**c**) Verify Score (Profiles-3D) of T1R1/T1R3.

**Figure 4 foods-14-04057-f004:**
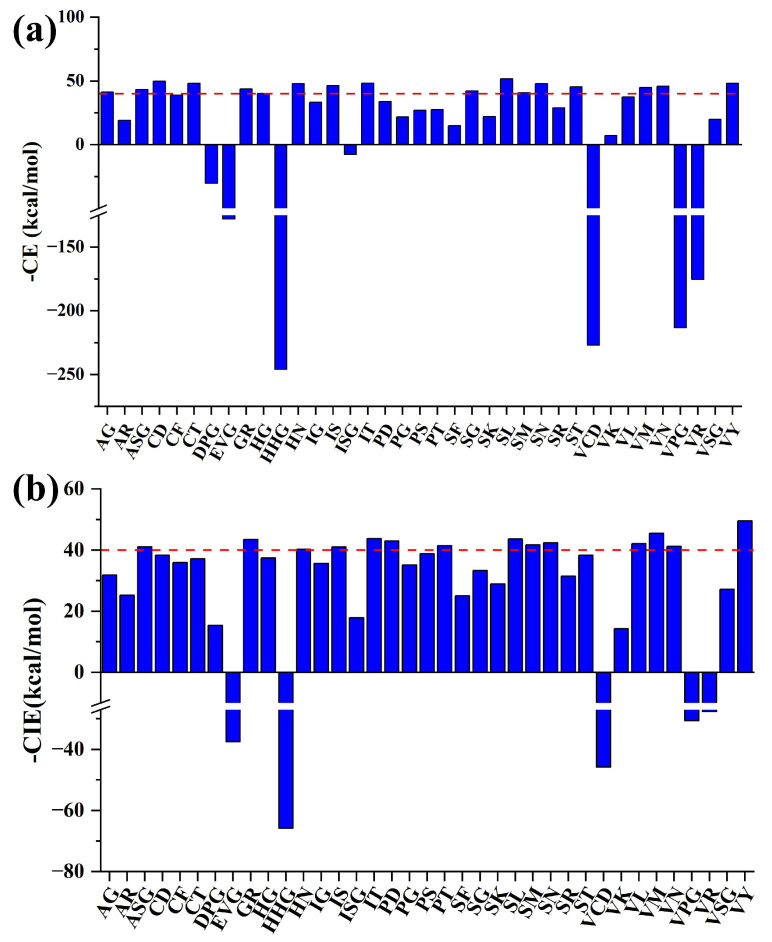
Molecular docking energy of peptides from yak bone collagen with T1R1/T1R3. (**a**,**b**), Unknown umami peptides with T1R1/T1R3 -CE (**a**) and -CIE (**b**), respectively. -CE and -CIE represent -CDOCKER energy and -CDOCKER interaction energy, respectively. The red dashed line represents the average docking scores between known umami peptides and T1R1/T1R3. CHSC, ICD, KPG, PRG, PSR, SHT, SPGE, SPT, and VCPE failed to dock with T1R1/T1R3.

**Figure 5 foods-14-04057-f005:**
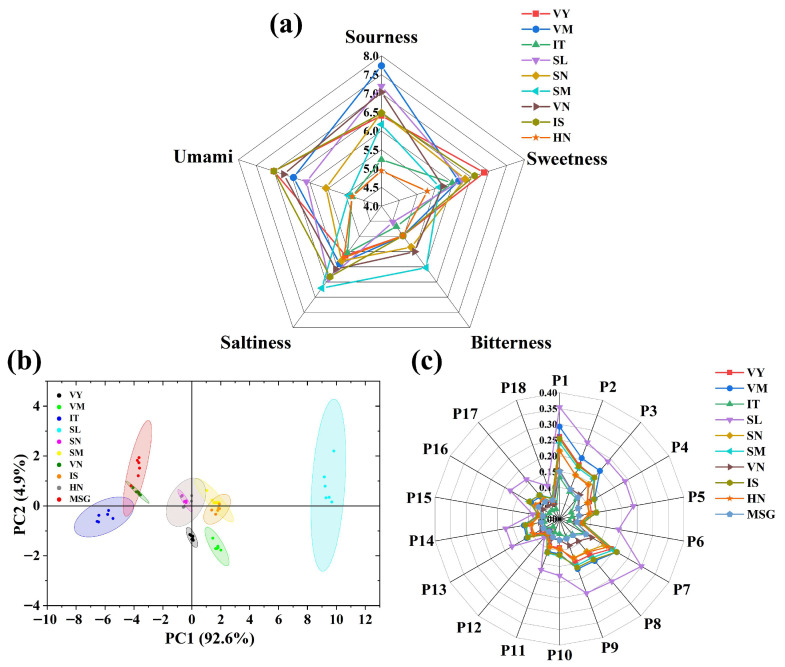
Sensory analysis of synthetic peptides. (**a**) Sensory evaluation results of synthetic peptides. (**b**) Electronic tongue principal component analysis of synthetic peptides. (**c**) Electronic tongue taste profile of synthetic peptides.

**Figure 6 foods-14-04057-f006:**
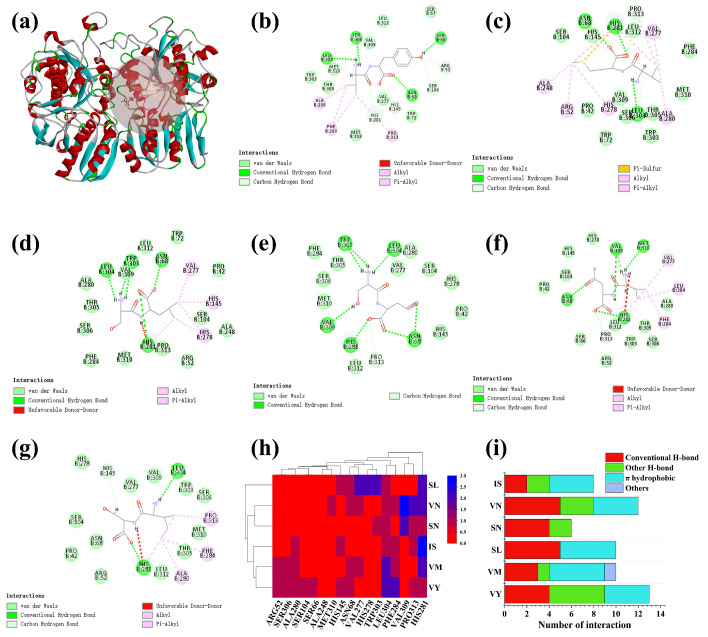
Underlying molecular mechanisms of identified yak collagen-derived peptides with T1R1/T1R3 revealed by molecular docking. (**a**) Docking site of the polypeptide with T1R1/T1R3. (**b**–**g**) 2D diagrams of VY (**b**), VM (**c**), SL (**d**), SN (**e**), VN (**f**), and IS (**g**) binding with the active center of T1R1/T1R3, respectively. (**h**) Heat map of the active site of the interaction of the umami peptide with T1R1/T1R3. (**i**) Statistics of non-bonding interaction modes for each identified peptide interacted with T1R1/T1R3.

**Figure 7 foods-14-04057-f007:**
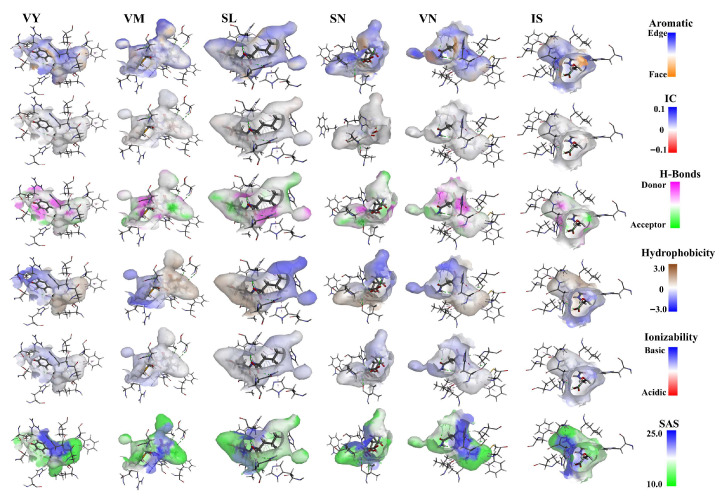
Surface force analysis of T1R1/T1R3 interacting with umami peptides.

**Table 1 foods-14-04057-t001:** *In silico* screening of potential umami peptides.

Peptide	Proportion of Umami Residues (%)	Water Solubility (log S)	Toxicity	Allergen
AHD	100.00	1.52	Non	Allergen
CCP	0.00	1.45	Non	Allergen
CSK	33.33	2.28	Non	Allergen
HPG	66.67	1.02	Non	Allergen
IPG	33.33	0.46	Non	Allergen
IPP	0.00	−0.30	Non	Allergen
ISK	33.33	1.36	Non	Allergen
NSG	66.67	3.19	Non	Allergen
PPSG	50.00	1.30	Non	Allergen
SCL	33.33	1.05	Non	Allergen
SHPE	50.00	1.83	Non	Allergen
SPG	66.67	2.52	Non	Allergen
VKN	33.33	1.56	Non	Allergen
VPC	33.33	0.75	Non	Allergen
VPT	66.67	1.10	Non	Allergen
VWY	66.67	−2.00	Non	Allergen
WSSG	75.00	0.92	Non	Allergen
AG	100.00	1.82	Non	Non
AR	50.00	2.37	Non	Non
CD	50.00	1.80	Non	Non
CF	50.00	0.32	Non	Non
CHSG	75.00	1.93	Non	Non
CK	0.00	1.14	Non	Non
CL	0.00	0.13	Non	Non
CN	0.00	2.22	Non	Non
CPN	0.00	2.35	Non	Non
CR	0.00	1.71	Non	Non
CT	50.00	1.67	Non	Non
DPG	66.67	2.29	Non	Non
EVG	100.00	1.80	Non	Non
HCK	33.33	1.10	Non	Non
HG	100.00	1.44	Non	Non
HHG	100.00	1.18	Non	Non
HK	50.00	1.22	Non	Non
HR	50.00	1.50	Non	Non
IC	0.00	2.22	Non	Non
ICD	33.33	0.99	Non	Non
IG	50.00	0.96	Non	Non
IK	0.00	1.27	Non	Non
IL	0.00	0.25	Non	Non
IN	0.00	2.10	Non	Non
IR	0.00	1.54	Non	Non
IS	50.00	1.38	Non	Non
ISG	66.67	1.63	Non	Non
IT	50.00	1.26	Non	Non
KPG	33.33	1.46	Non	Non
LK	0.00	1.11	Non	Non
LR	0.00	1.37	Non	Non
PD	50.00	2.03	Non	Non
PG	50.00	1.76	Non	Non
PM	0.00	1.04	Non	Non
PRG	33.33	1.50	Non	Non
PS	50.00	2.25	Non	Non
PSR	33.33	2.67	Non	Non
SF	100.00	1.03	Non	Non
SG	100.00	2.45	Non	Non
SHT	100.00	2.01	Non	Non
SK	50.00	1.86	Non	Non
SL	50.00	0.84	Non	Non
SM	50.00	1.22	Non	Non
SN	50.00	2.93	Non	Non
SPGE	75.00	2.59	Non	Non
SPT	66.67	2.51	Non	Non
SR	50.00	2.12	Non	Non
ST	100.00	2.38	Non	Non
VCD	66.67	1.01	Non	Non
VCF	66.67	−0.48	Non	Non
VCPE	50.00	1.33	Non	Non
VK	50.00	1.52	Non	Non
VL	50.00	0.50	Non	Non
VM	50.00	0.88	Non	Non
VN	50.00	2.53	Non	Non
VPG	66.67	1.17	Non	Non
VR	50.00	1.78	Non	Non
VSG	100.00	2.27	Non	Non
VW	50.00	−0.77	Non	Non
VWK	33.33	−0.81	Non	Non
VY	100.00	0.25	Non	Non
WG	50.00	−0.11	Non	Non
WY	50.00	−1.29	Non	Non
HN	50.00	1.98	Non	Non
PT	50.00	1.89	Non	Non

Note: log S, the logarithmic value of solubility (M) in water.

**Table 2 foods-14-04057-t002:** Physicochemical properties and biological potential for candidate peptides by *in silico* analysis.

Peptide	Mass	pI	Net Charge	Hydrophobicity (kcal/mol)	HIA Probability	HOB (%)
IS	218.1263	5.46	0	7.24	1	88
IT	232.1419	5.36	0	7.03	1	83
SL	218.1263	5.50	0	7.11	1	87
SM	236.0828	5.33	0	7.69	1	82
SN	219.0853	5.34	0	9.21	1	77
VM	248.1191	5.40	0	6.77	1	91
VN	231.1216	5.41	0	8.29	1	86
VY	280.1419	5.45	0	6.73	1	74
HN	269.1122	7.69	0	11.08	1	61

Note: pI, isoelectric point; HIA, human intestinal absorption; HOB, human oral bioavailability.

## Data Availability

The original contributions presented in the study are included in the article/Appendix A, further inquiries can be directed to the corresponding authors.

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
