# Peer review of "Foods2025, 14(23), 4057;https://doi.org/10.3390/foods14234057"

_foods, 2025, doi:10.3390/foods14234057_

Round 1

Reviewer 1 Report

Comments and Suggestions for Authors

The review article “Identification of novel umami peptides from yak bone collagen and mechanism exploration through in silico discovery, molecular docking, and electronic tongue” by Mei and coworkers presents a comprehensive evaluation of the umami peptides from their theoretical analysis to synthesis and electronic tongue examination. Articles included in the reference list are up-to-date and cover the most important aspects. The authors have presented results in a systematic manner and different sections are well-connected. The article may be of interest to Foods readers, although some points require further consideration before a final decision is made. Therefore, I recommend a major revision.

The authors should answer the following:

  1. Figure 1 should be moved to the Supplementary information, as it is not very informative. It would be better to present the structure of the modeled protein, with the reference to the changes in structure compared to the bovine bone collagen
  2. The authors should mention some of the known limitations of the prediction of physicochemical properties by the in silico analysis.
  3. The molecular docking energies shown in a graph are not very informative, as the x axis is not clearly visible, the authors should present the values in Table 1
  4. The authors should give more details about the docking to T1R1/T1R3 – how was the active pocket determined, which of the amino acids are involved in binding, what kind of interactions are formed.
  5. The authors should give a chemical rationale for the strong binding of specific peptides to the receptor, which functional groups are responsible for the interactions. The structures of the peptides that bind strongly to the receptor should be presented.
  6. The authors should specify how the space geometry of the peptides was determined and if these can used to reliably present the confirmations of the peptides. Citing the appropriate literature is needed.
  7. The authors mention the unknown umami peptides, citing the appropriate literature is needed to verify the chemical rationale for their existence.
  8. Figure 7 should be enlarged as it is not clear.
  9. The authors should present spectroscopic data related to the synthesis of umami peptide candidates, how was their structure determined?
  10. The electronic tongue results should be compared to the literature data

Author Response

Comment 1: Figure 1 should be moved to the Supplementary information, as it is not very informative. It would be better to present the structure of the modeled protein, with the reference to the changes in structure compared to the bovine bone collagen.

Response 1: Thank you for reviewing our manuscript and providing valuable feedback. Regarding your suggestion to move Figure 1 to the Supplementary Materials and replace it with a three-dimensional structure diagram of the modeled protein , we have carefully considered this and believe it is more appropriate to retain Figure 1 at this time for the following reasons:

  1. Given that neither the UniProt nor NCBI databases contain experimentally determined structures for bovine (Bos taurus) or yak (Bos grunniens) collagen, prediction is currently only possible through computational methods. However, structural prediction is not the core focus of this study and bears little relevance to the present work.
  2. Subsequent analyses in this paper are conducted at the sequence level and do not involve structural information (except for molecular docking). Therefore, replacing Figure 1 with a three-dimensional structural diagram would neither provide additional support for the main text nor serve to distract readers.

Therefore, we have retained the sequence alignment and believe it most effectively serves the purpose of illustrating the foundational sequence-level relationship between our modeled protein and the bovine bone collagen template (page 6, lines 239–240).

Comment 2: The authors should mention some of the known limitations of the prediction of physicochemical properties by the in silico analysis.

Response 2: We greatly appreciate your valuable suggestions regarding our article. Based on your recommendations, we have added the following content in the revised version : In silico prediction accelerates peptide activity screening through its low-cost, high-throughput advantages, yet remains constrained by issues such as data quality, model interpretability, and insufficient experimental validation. Its practical application still requires breakthroughs (page 15, lines 400–403).

Comment 3: The molecular docking energies shown in a graph are not very informative, as the x axis is not clearly visible, the authors should present the values in Table 1

Response 3: Thank you for your thorough review and valuable feedback. We have regenerated Figure 4 at 600 dpi and enlarged the axis labels, so the bar chart and x-axis labels are now clearly visible. Furthermore, we believe converting the data in Figure 4 into a table would diminish the intuitive comparative advantage provided by the bar chart. (page 13, line 381) Therefore, we prefer to retain Figure 4 for visual comparison rather than moving it to Table 1. In accordance with your suggestion, we have submitted the docking energy data for known peptide segments to the Supplementary Materials (Table S1) (page 10, lines 364–370).

Comment 4: The authors should give more details about the docking to T1R1/T1R3 – how was the active pocket determined, which of the amino acids are involved in binding, what kind of interactions are formed.

Response 4: Thank you for reminding us to further supplement the docking details.

Firstly, Regarding the positioning of the active site pocket, the original manuscript already explained that it referenced the mechanism of VFT domain closure following umami peptide binding to T1R1/T1R3 (page 2, lines 67-70) and drew upon previously reported T1R3-VFT binding sites (page 4, lines 154-155). 

Secondly, the amino acid residues at the binding sites involved in the T1R1/T1R3 docking interface are detailed in Figures 6h and 6i. As per your suggestion, we have supplemented the corresponding explanation in the main text: Hydrogen bonds form far more often than other kinds of interactions, such as ASN68, LEU304, SER306, SER66, HIS281, TRP303, VAL309, MET310, ARG52, SER104, HIS145, and PRO313. These residues participate in the interactions of most identified peptides via hydrogen bonds, and this matches existing findings that highlight hydrogen bonds as vital for umami peptides to bind stably to T1R1/T1R3 receptors. Furthermore, the hydrophobic interactions formed between peptide chains and residues(Including ALA280, PRO313, PHE284, VAL277, LEU304, ALA280, ARG52, HIS281, HIS145, and HIS278) of the T1R1/T1R3 taste receptors effectively boost the binding strength between the two, which in turn makes taste perception stronger (Page 16, lines 477-485).

Comment 5: The authors should give a chemical rationale for the strong binding of specific peptides to the receptor, which functional groups are responsible for the interactions. The structures of the peptides that bind strongly to the receptor should be presented.

Response 5: we thank the reviewer for this insightful suggestion.

Figures 6b-g clearly illustrate the 2D interaction map between the peptide and T1R1/T1R3, revealing in detail the functional groups driving strong binding—such as the hydroxyl group of the tyrosine side chain in peptide VY forming a hydrogen bond with SER66. Furthermore, we have significantly expanded the discussion on the molecular interactions responsible for the strong binding of the peptides to the T1R1/T1R3 receptor: Hydrogen bonds form far more often than other kinds of interactions, such as ASN68, LEU304, SER306, SER66, HIS281, TRP303, VAL309, MET310, ARG52, SER104, HIS145, and PRO313. These residues participate in the interactions of most identified peptides via hydrogen bonds, and this matches existing findings that highlight hydrogen bonds as vital for umami peptides to bind stably to T1R1/T1R3 receptors [17]. Furthermore, the hydrophobic interactions formed between peptide chains and residues(Including ALA280, PRO313, PHE284, VAL277, LEU304, ALA280, ARG52, HIS281, HIS145, and HIS278) of the T1R1/T1R3 taste receptors effectively boost the binding strength between the two, which in turn makes taste perception stronger(Page 16, lines 477-485).

Comment 6: The authors should specify how the space geometry of the peptides was determined and if these can used to reliably present the confirmations of the peptides. Citing the appropriate literature is needed.

Response 6: Thank you very much for this helpful comment. We have now added the following clarification and references to the revised manuscript :  all potential umami peptides were constructed using the “Protein Preparation” module in DS2019 software and underwent preprocessing such as energy minimization, followed by semi-flexible docking via the CDOCKER protocol(page 4, lines 155–157).

Comment 7: The authors mention the unknown umami peptides, citing the appropriate literature is needed to verify the chemical rationale for their existence.

Response 7: Thank you very much for your valuable comments. Peptides previously documented in the BIOPEP-UWM database or literature are defined as known umami peptides; those not found in any public database or literature are classified as unknown. All unknown peptides reported in this paper belong to the unknown category, hence there are no prior references to cite (page 9, lines 311–322).

Comment 8: Figure 7 should be enlarged as it is not clear.

Response 8: Thank you very much for pointing out the issue with Figure 7. We have now enlarged the figure and re-exported it at 600 dpi, which significantly improves its clarity and readability. We appreciate your attention to detail and hope the revised figure meets your expectations (page 18, lines 511).

Comment 9: The authors should present spectroscopic data related to the synthesis of umami peptide candidates, how was their structure determined?

Response 9: We appreciate the reviewer's comment regarding the structural confirmation of the umami peptide candidates.

Firstly, in our study, the peptides were obtained commercially through custom synthesis for bioactivity testing. The primary documentation provided by the manufacturer is the High-Performance Liquid Chromatography (HPLC) and Mass Spectrometry (MS) analysis report, which we have now provided as Supplementary Figure S1 in the revised manuscript and cited in the main text (Page 10, Line 370-372). This report confirms the high purity of the peptides, which is the most critical parameter for ensuring the reliability of our subsequent sensory evaluation experiments.

Comment 10: The electronic tongue results should be compared to the literature data

Response 10: We sincerely appreciate your valuable feedback. Based on your suggestions, we have made the following corrections to the previous manuscript: Furthermore, research by Yu et al. [58] indicates that discrepancies also exist between the results obtained by electronic tongues and sensory evaluations (page 15, lines 439-441).

Reviewer 2 Report

Comments and Suggestions for Authors

The authors present an interesting study aimed at identifying novel umami peptides from yak bone collagen using an in silico approach, which reduces costs. However, there are some points that should be clarified:

1) It is not clearly explained how the number of bioactive fragments in a sequence (aumami) is counted. Is this related to the presence of any of the 10 umami amino acids identified by Liu [48] and Zhang [49]?

2) How were the interactions obtained? Did the authors use any specific package within Discovery Studio for this purpose?

3) The authors should consider using the same scale for the y-axis in both plots of Figure 4. Moreover, the x-axis labeling should be improved to facilitate peptide identification.

4) One of the criteria for selecting peptides for further studies was that the docking values of -CE and -CIE should be greater than 40 kcal/mol. Why did the authors choose this particular cutoff value?

5) Regarding docking values, the authors frequently use the negative forms of both CE (-CE) and CIE (-CIE) to express binding affinity, but the “direct” CDOCKER energy terms are also used. These inconsistencies in how the terms are applied throughout the paper may cause confusion. See, for example: Section 2.5, line 158; Section 3.4, line 314; and Section 3.8, lines 428–430. The authors should ensure consistency in how energy values are expressed.

Author Response

Comment 1: It is not clearly explained how the number of bioactive fragments in a sequence (Aumami) is counted. Is this related to the presence of any of the 10 umami amino acids identified by Liu [48] and Zhang [49]?

Response 1: Thank you very much for your valuable review and comment. We are sorry for not explaining the counting rule more clearly. Actually, Aumami = aumami/N, where aumami was the number of umami fragments in the selected sequence included in the BIOPEP-UWM database, and N was the number of total amino acid residues in this sequence. There was not any relevance between the calculation of Aumami or DH and the report of Liu [48] and Zhang [49]. According to your comment, the definitions of Aumami, DH, aumami, N, d, and D have been modified to be clearer in the revised manuscript (page 3, lines 127-129).

A short statement was also added in the revised manuscript: “The data of aumami, N, d, and D were all sourced from the BIOPEP-UWM database” (page 3, lines 131-132).

Comment 2: How were the interactions obtained? Did the authors use any specific package within Discovery Studio for this purpose?

Response 2: Thank you for your valuable review and comment. All intermolecular interactions were generated automatically by the standard tool embedded in Discovery Studio 2019 software, and there was not any additional specific package used in this analysis (page 4, lines 163-165).

Comment 3: The authors should consider using the same scale for the y-axis in both plots of Figure 4. Moreover, the x-axis labeling should be improved to facilitate peptide identification.

Response 3: Thank you for your valuable suggestion. According to your suggestion, we have re-plotted Figure 4 at a precision of 600 dpi in the revised manuscript, and the x-axis labeling was also improved to allow the peptide identification clearer and easier (page 13, line 381).

Comment 4: One of the criteria for selecting peptides for further studies was that the docking values of -CE and -CIE should be greater than 40 kcal/mol. Why did the authors choose this particular cutoff value?

Response 4: Thank you for your valuable review and comment. Actually, we also performed the molecular docking between the 14 previously reported umami peptides within the released fragments (Figure 2e-f) and T1R1/T1R3, and the results showed that the average -CIE and -CE values were 40.92 and 43.03 kcal/mol, respectively. Hence, we have chosen >40 kcal/mol as the cutoff value for selecting novel umami peptide candidates. According to your valuable comment, this explanation has been added in the revised manuscript (page 10, lines 364-370) and the above-mentioned data were also added in the supplementary file (Table S1).

Comment 5: Regarding docking values, the authors frequently use the negative forms of both CE (-CE) and CIE (-CIE) to express binding affinity, but the “direct” CDOCKER energy terms are also used. These inconsistencies in how the terms are applied throughout the paper may cause confusion. See, for example: Section 2.5, line 158; Section 3.4, line 314; and Section 3.8, lines 428–430. The authors should ensure consistency in how energy values are expressed.

Response 5: Thank you very much for your valuable suggestion. According to your suggestion, all the “-CDOCKER energy” and “-CDOCKER interaction energy” terms have been modified to the “-CE” and “-CIE” terms, respectively, across the whole context of the revised manuscript (page 9, line328; page 15, line 457). Additionally, we have added a full explanation of the abbreviation “-CDOCKER interaction energy (-CIE)” and “-CDOCKER energy (-CE)” upon its first appearance in the revised manuscript (page 4, lines 160-161).

Reviewer 3 Report

Comments and Suggestions for Authors

The study focuses on obtaining target peptides from enzymatic protein hydrolysates. In this case, the authors examine umami-flavored peptides from yak bone collagen hydrolyzed by a number of proteases and their blends. Proteolysis followed by mass-spectrometry identification is omitted, and possible peptide fragments are predicted using existing computer algorithms. A key advantage of this study is the computer modeling of peptide interactions with receptors T1R1/T1R3. However, it turns out that this modeling is insufficient for predicting umami flavor.

I think that the article can be improved considering the following comments and suggestions:

- The authors use a computer prediction of potentially released peptide fragments and do not support this with an experiment, i.e., identification of peptides in real enzymatic hydrolysates. When discussing their results, the authors are encouraged to provide references showing the percentage agreement between the predicted peptides and experimental data. To understand the capabilities of the authors' method, it is necessary to present this quantitative assessment, possibly obtained for other protein substrates and other enzymes.

- There is no need to provide 5 or even 6 significant digits for the DH (lines 256-259).Experimental DH values ​​are determined with an accuracy of 3-5%.It is sufficient to provide 3 significant digits, since more precise calculated values simply have nothing to compare with.

- In Figure 2, it is necessary to indicate which symbols belong to DH and Aumami.

- The authors write “Aumami values exhibited a pronounced positive linear relationship with DH values” (lines 250 and 251). It is difficult to understand this from the graph. Authors need to provide the slope and correlation coefficient.

- The authors report high degrees of hydrolysis for pepsin (>60%). They should cite experimental studies on pepsin hydrolysis of other protein substrates and discuss the degrees of hydrolysis obtained in these studies with their own theoretical results.

Author Response

Comment 1: The authors use a computer prediction of potentially released peptide fragments and do not support this with an experiment, i.e., identification of peptides in real enzymatic hydrolysates. When discussing their results, the authors are encouraged to provide references showing the percentage agreement between the predicted peptides and experimental data. To understand the capabilities of the authors' method, it is necessary to present this quantitative assessment, possibly obtained for other protein substrates and other enzymes.

Response 1: Thank you very much for your careful review and valuable suggestion. According to your valuable suggestion, the capability of in silico simulated proteolysis has been compared and discussed with the practical proteolytic method based on the previously peer-reviewed reports [1,2]. Huang et al. [30] found that two xanthine oxidase inhibitory peptides identified from fire hemp seed hydrolysates contain three released peptide sequences through in silico simulated enzymolysis. Gao et al. [31] reported that the key results from in silico and in vitro simulated gastrointestinal digestion are highly consistent, particularly demon-strating a complete alignment in functional activity trends and molecular mechanism validation. These statements have also been added in the revised manuscript (page 7, lines 245-252).

References:

[1] Guo, D.; Zou, H.; Chen, M.; Wei, S.; Cai, Y.; Wang, Z.; Wang, H.; Yi, Y.; Xu, W. Corn-derived peptide LQQQLL alleviates skeletal muscle attenuation by mTOR signaling pathway and intestinal microbiota. Food Research International 2025, 218, 116872, doi:https://doi.org/10.1016/j.foodres.2025.116872.

[2] Gao, R.; Kong, F.; Mu, G.; Zhao, X.; Cheng, J.; Qian, F. Screening of enzymes for bi-functional whey protein hydrolysates: virtual enzymolysis, fragmentomics, and molecular docking in silico. Food Research International 2025, 214, 116629, doi:https://doi.org/10.1016/j.foodres.2025.116629.

Comment 2: There is no need to provide 5 or even 6 significant digits for the DH (lines 256-259). Experimental DH values are determined with an accuracy of 3-5%. It is sufficient to provide 3 significant digits, since more precise calculated values simply have nothing to compare with.

Response 2: Thank you very much for your careful review and suggestion. According to your valuable suggestion, all the DH values have been presented as the 3-digit numbers in the revised manuscript (page7, lines 266-270; page 8, line 294).

Comment 3: In Figure 2, it is necessary to indicate which symbols belong to DH and Aumami.

Response 3: Thank you for your careful review and comment. Actually, in this figure, the Aumami values were presented as columns, while the DH values were exhibited as scatters. Following your valuable suggestion, this information has been added in the figure captions in the revised manuscript (page 8, lines 301-302).

Comment 4: The authors write “Aumami values exhibited a pronounced positive linear relationship with DH values” (lines 250 and 251). It is difficult to understand this from the graph. Authors need to provide the slope and correlation coefficient.

Response 4: Thank you for your careful review and valuable comment. We are sorry for the incorrect description of "pronounced positive linear relationship". Actually, it just showed an overall trend of positively correlated, which only means that a higher theoretical DH, a probably higher Aumami value. Therefore, according to your valuable comment, this statement has been modified to “The Aumami value seems overall positively correlated with the theoretical DH value” in the revised manuscript (page 7, line 261).

Comment 5: The authors report high degrees of hydrolysis for pepsin (>60%). They should cite experimental studies on pepsin hydrolysis of other protein substrates and discuss the degrees of hydrolysis obtained in these studies with their own theoretical results.

Response 5: Thank you for your valuable comment and suggestion. According to your valuable suggestion, our above finding was further compared with a previously peer-reviewed report of Zhang et al. [1]. Zhang et al. [1] found that the Huangjiu lee hydrolysates from pepsin enzymolysis exhibited significantly higher total peptide yield than those hydrolyzed by alkaline protease, neutral protease, and papain, which agreed with our above findings. This information has also been added in the revised manuscript (page 7, line 270-272).

References:

[1] Zhang, R.; Zhou, Z.; Ji, Z.; Ren, Q.; Liu, S.; Xu, Y.; Mao, J. Identification and characterization of novel dual-function anti-oxidant and umami peptides from protein hydrolysates of Huangjiu lees. Food Quality and Safety 2024, 8, fyae011, doi:10.1093/fqsafe/fyae011.

Reviewer 4 Report

Comments and Suggestions for Authors

Dear Authors,

Please find my suggestions below.

In the abstract are missing the values of the most important results. Insert it in abstract.

Is there any specific geographic origin of yak bone collagen? Explain it in the manuscript.

Highlight more in the discussion practical importance of the peptide studied in this manuscript.

What are advancements of the peptides studied in this manuscript in comparison to the compounds which are commonly used today. Explain it in the manuscript.

Figure 6 is not visible well. Improve its quality.

Author Response

Comment 1: In the abstract are missing the values of the most important results. Insert it in abstract.

Response 1: Thank you for your careful review and valuable comment. According to your valuable comment, some important values have been added in the Abstract of the revised manuscript (page 1, lines 17-34), such as the number of proteases used in the simulated proteolysis, the binding affinity between selected peptide candidates and the T1R1/T1R3 receptor, the taste perception level of finally identified umami peptides, etc.

Comment 2: Is there any specific geographic origin of yak bone collagen? Explain it in the manuscript.

Response 2: Thank you for your valuable comment. Actually, yak is a characteristic animal mainly surviving in the Qinghai-Tibetan Plateau of Western China with an altitude >3000 m and an oxygen content <75% (in a plain basis). According to your comment, this information about specific geographic origin of yak has been added in the revised manuscript (page 1, lines 39-42).

Comment 3: Highlight more in the discussion practical importance of the peptide studied in this manuscript.

Response 3: Thank you for your careful review and valuable suggestion. Actually, monosodium glutamate (MSG) remains the most commonly used umami additive. However, due to its high sodium content, an excessive consumption of MSG may lead to hypertension, hyperuricemia, and other cardiovascular diseases. Studies showed that some free L-amino acids and peptides can deliver or intensify the favorable umami perception. The emergence of umami peptides, particularly the bone-derived umami peptides studied in this manuscript, could perfectly address the shortcoming of MSG, and thereby they have become a prospective alternative of MSG applied in food industry. According to your valuable suggestion, these statements that heighted the practical importance of the peptide studied have been added in the revised manuscript (page 2, lines 58-64).

Comment 4: What are advancements of the peptides studied in this manuscript in comparison to the compounds which are commonly used today. Explain it in the manuscript.

Response 4: Thank you very much for your valuable suggestions. Actually, the most commonly used umami additive is monosodium glutamate (MSG), but an excessive consumption of MSG may lead to hypertension, hyperuricemia, and other cardiovascular diseases due to its high sodium content. The emergence of umami peptides, particularly the bone-derived umami peptides studied in this manuscript, could perfectly address the shortcoming of MSG, and thereby they have become a prospective alternative of MSG applied in food industry. According to your valuable suggestion, these statements about the advancements of the peptide studied in comparison to the commonly compound MSG have been added in the revised manuscript (page 2, lines 58-64).

Comment 5: Figure 6 is not visible well. Improve its quality.

Response 5: Thank you very much for your careful review and comment. According to your valuable suggestion, we have re-plotted Figure 6 at a precision of 600 dpi and enlarged its core areas in the revised manuscript (page 17, line 502).

Round 2

Reviewer 1 Report

Comments and Suggestions for Authors

The authors have answered all of the questions properly and improved the manuscript. The manuscript is suitable for publication in the present form. 

Reviewer 4 Report

Comments and Suggestions for Authors

Regarding the second review manuscript is fine for me. All suggestions were accepted and manuscript is improved. It can be accepted.